# Elastic Photon Scattering on Hydrogenic Atoms near Resonances

**Dmitrii Samoilenko** [1,2,3,*], **Andrey V. Volotka** [1,2]  **and Stephan Fritzsche** [1,2,3]

1. Helmholtz-Institut Jena, D-07743 Jena, Germany; a.volotka@gsi.de (A.V.V.); s.fritzsche@gsi.de (S.F.)
2. GSI Helmholtzzentrum für Schwerionenforschung, D-64291 Darmstadt, Germany
3. Theoretisch-Physikalisches Institut, Friedrich-Schiller-Universität, D-07743 Jena, Germany
* Correspondence: dmitrii.samoilenko@uni-jena.de

**Abstract:** Scattering of light on relativistic heavy ion beams is widely used for characterizing and tuning the properties of both the light and the ion beam. Its elastic component—Rayleigh scattering—is investigated in this work for photon energies close to certain electronic transitions because of its potential usage in the Gamma Factory initiative at CERN. The angle-differential cross-section, as well as the degree of polarization of the scattered light are investigated for the cases of $1s - 2p_{1/2}$ and $1s - 2p_{3/2}$ resonance transitions in H-like lead ions. In order to gauge the validity and uncertainty of frequently used approximations, we compare different methods. In particular, rigorous quantum electrodynamics calculations are compared with the resonant electric-dipole approximation evaluated within the relativistic and nonrelativistic formalisms. For better understanding of the origin of the approximation, the commonly used theoretical approach is explained here in detail. We find that in most cases, the nonrelativistic resonant electric-dipole approximation fails to describe the properties of the scattered light. At the same time, its relativistic variant agrees with the rigorous treatment within a level of 10% to 20%. These findings are essential for the design of an experimental setup exploiting the scattering process, as well as for the determination of the scattered light properties.

**Keywords:** Rayleigh scattering; fluorescence spectroscopy; polarization transfer

## 1. Introduction

Absorption and emission of a photon by an atom are the fundamental processes in atomic physics, which are of utmost importance for the characterization of the atomic system. In particular, the laser-induced fluorescence spectroscopy of atoms and molecules is based on an electronic excitation and subsequent fluorescence, which defines the electronic structure of a sample [1]. The energy scale of the fluorescence spectroscopy is typically located between infrared and ultraviolet regions. In order to describe the electronic structure of atoms and ions with binding energies in this energy range, it is often enough to employ the nonrelativistic theory. However, as the interest of atomic physics spreads to a higher energy domain, experimental findings reveal that in some cases, even relativistic consideration might not be enough. One of the most common examples of such a situation is the consideration of core (deeply bound) electrons in heavy atoms or ions. A proper description of the spectral properties involving the strongly bound electrons requires the incorporation of quantum electrodynamics (QED) effects [2]. Both relativistic and QED considerations can be, however, very tedious and demand from the researcher a background in the corresponding fields and sometimes considerable computational resources. At the same time, the well-known and easy-to-use Bohr theory based method of the description of the electronic structure remains attractive in its simplicity. It can provide preliminary estimations even for modern applications. The necessity of more complex considerations is, of course, to be decided in every particular case individually.

Here, we consider the photon absorption and emission (fluorescence spectroscopy) of H-like lead ion $Pb^{81+}$ in its ground state with photon energy being close (but not exact) to $1s - 2p_{1/2}$ and $1s - 2p_{3/2}$ resonances. Ab initio, the problem can be formulated in terms of elastic photon scattering or Rayleigh scattering. First, the incident photon is absorbed by an ion, and the ion undergoes the transition from the initial state into an intermediate virtual (or real) state. After that, the ion decays to the final state and emits the scattered photon. Rayleigh scattering is an important and widely studied process in atomic physics. Being one of the basic second-order processes of QED, Rayleigh scattering is extensively studied theoretically [3–11], as well as experimentally [12–14]. Apart from the fundamental interest, Rayleigh scattering is actively utilized in various fields, including the study of the solid-state, complex molecules, or nano-objects [15–18], astrophysics [19,20], as well as in medical diagnostics [21].

In this work, we investigate the angular distribution of two observables: cross-section and polarization of the scattered light. We carry out the calculations with three different approaches and compare the obtained results. The simplest approximation we refer to as the nonrelativistic resonant electric dipole one. Electron binding energies and wave functions are obtained by solving the ordinary Schrodinger equation; as an intermediate state, only the nearest resonant level is taken into account; and the electric dipole operator describes the photon-ion interaction. The second approach is the relativistic resonant electric dipole approximation, wherein wave functions and binding energies are the solutions of the Dirac equation. However, we still assume only one (resonant) intermediate state and electric dipole transition operator. Finally, the third consideration we refer to as exact. Here, we add to the relativistic resonant electric dipole approximation two things: (i) sum over all possible intermediate states and (ii) include all terms in the multipole expansion of the transition operator. We believe that the obtained results can be helpful for many applications that exploit the near-resonant Rayleigh scattering. In particular, our example focuses on the Gamma Factory initiative [22]. This project is pursuing building at CERN (European Organization for Nuclear Research) a $\gamma$-ray source with photon energies several orders of magnitude higher than available currently anywhere in the world with beam intensities comparable with those produced by free-electron lasers. The idea is to store relativistic highly charged heavy ions at CERN's Large Hadron Collider (LHC) and scatter optical laser light on these ions. If we choose a laser wavelength to match (keeping in mind the relativistic Doppler effect) a certain electronic transition in the ions, the scattering cross-section (and, consequently, the output intensity) grows drastically. With values of the Lorentz factor $\gamma \approx 3000$, feasible at LHC, Doppler-enhanced optical frequencies correspond to $1s \rightarrow 2p$ transitions in elements from the lead region of the periodic table. Recently, there have been the first successful proof-of-principle experiments with $Pb^{81+}$ ions [23,24], produced with Al stripper foil [25]. For that reason, we have chosen lead among other high-Z elements for consideration.

## 2. Materials and Methods

### 2.1. Statement of the Problem

The process under consideration can be illustrated as follows:

$$|i\rangle + \gamma_i \rightarrow |a\rangle \rightarrow |f\rangle + \gamma_f, \tag{1}$$

where the incident photon $\gamma_i$ with polarization vector $\epsilon_i$ and momentum $\mathbf{k}_i$ is absorbed by an ion, and the ion is brought from the initial state $|i\rangle = |1s\rangle$ to the intermediate state $|a\rangle$, which in the resonant case is restricted to $2p_{1/2}$ or $2p_{3/2}$ states only. After that, the ion decays to the final state $|f\rangle = |1s\rangle$ and emits the scattered photon $\gamma_f$ with momentum $\mathbf{k}_f$ and polarization vector $\epsilon_f$. The geometry of the described process is shown in Figure 1. The ion is assumed to be located at the origin of the coordinate system. The z-axis in the Cartesian coordinates is oriented along $\mathbf{k}_i$, and the x-axis is chosen in the direction of $\epsilon_i$ if the light is linearly polarized. The scattered photon propagates in the direction of $\mathbf{k}_f$. Polar angle $\vartheta$ is measured between the propagation directions

of the initial and the scattered photons and referred to as the scattering angle. At the same time, the azimuthal angle $\varphi$ is the angle between the $x$-axis and the projection of $\mathbf{k}_f$ on the $xy$-plane.

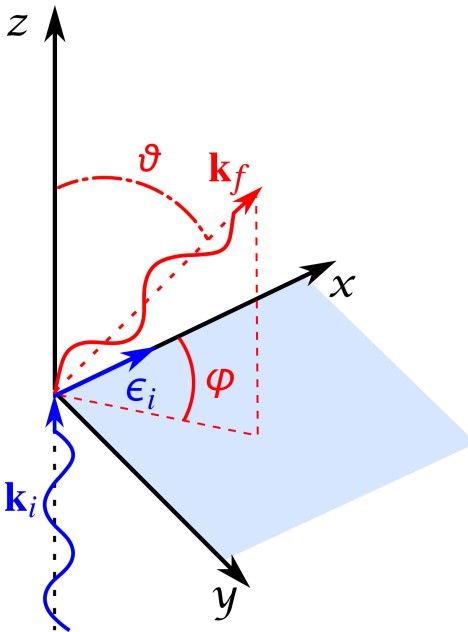

**Figure 1.** (Color online) The adopted geometry for the scattering process. The $z$-axis is chosen along the direction of the momentum of the incident photon $\mathbf{k}_i$, while the $x$-axis is fixed by its polarization vector $\boldsymbol{\epsilon}_i$. The polar (scattering) $\vartheta$ and azimuthal $\varphi$ angles uniquely define the direction of the scattered photon $\mathbf{k}_f$.

The problem can be conveniently formulated in terms of the density matrix formalism. Density operators of the incident ($\hat{\rho}_i$) and the scattered (final) ($\hat{\rho}_f$) photons are connected by a well-known relation [26]:

$$\hat{\rho}_f = \hat{\mathcal{M}} \hat{\rho}_i \hat{\mathcal{M}}^\dagger \,, \tag{2}$$

with $\hat{\mathcal{M}}$ being a scattering operator that characterizes the scattering process. The explicit expressions for this operator are given in the following subsection. Since the incident and scattered photons have well defined wave vectors, we write the photon density matrices in the helicity representation as follows:

$$\langle \mathbf{k}\lambda | \hat{\rho} | \mathbf{k}\lambda' \rangle = \frac{I}{2} \begin{pmatrix} 1 + P_3 & P_1 - iP_2 \\ P_1 + iP_2 & 1 - P_3 \end{pmatrix} \,, \tag{3}$$

where $I$ stands for the intensity of the light and the Stokes parameters $P_{1,2}$ and $P_3$ define the degree of linear and circular polarization, respectively. For the scattering cross-section calculations, we assume the normalized incident light, i.e., its intensity is equaled to one. Keeping this in mind, the angle-differential cross-section is given by the trace of the density matrix of the scattered photon [26]:

$$\frac{d\sigma}{d\Omega} = \langle \mathbf{k}_f - 1 | \hat{\rho}_f | \mathbf{k}_f - 1 \rangle + \langle \mathbf{k}_f + 1 | \hat{\rho}_f | \mathbf{k}_f + 1 \rangle \,. \tag{4}$$

In turn, the degree of linear polarization of the scattered light $P_l$ can be found as:

$$P_l = \sqrt{P_1^2 + P_2^2} \,. \tag{5}$$

Now, let us turn to the details of the calculations.

## 2.2. General Theory of Transition Matrix Amplitude

We adopt the Furry picture for our calculations. That means that the electron-nucleus interaction is incorporated in the unperturbed Hamiltonian of the system and the ion-photon interaction is considered as perturbation. For the determination of the properties of the scattered light, we then evaluate the second-order scattering operator $\hat{\mathcal{M}}$, the matrix elements of which are defined as (in units $\hbar = m_e = c = 1$) [11,27]:

$$\mathcal{M}(m_i, m_f, \lambda_i, \lambda_f) = \frac{\alpha}{\omega} \sum_a \left( \frac{\langle f|\boldsymbol{\alpha} \cdot \boldsymbol{\epsilon}^*_{\lambda_f} e^{-i\mathbf{k}_f \cdot \mathbf{r}}|a\rangle \langle a|\boldsymbol{\alpha} \cdot \boldsymbol{\epsilon}_{\lambda_i} e^{i\mathbf{k}_i \cdot \mathbf{r}}|i\rangle}{E_i + \omega - E_a} + \frac{\langle f|\boldsymbol{\alpha} \cdot \boldsymbol{\epsilon}_{\lambda_i} e^{i\mathbf{k}_i \cdot \mathbf{r}}|a\rangle \langle a|\boldsymbol{\alpha} \cdot \boldsymbol{\epsilon}^*_{\lambda_f} e^{-i\mathbf{k}_f \cdot \mathbf{r}}|i\rangle}{E_f - \omega - E_a} \right), \tag{6}$$

where $\alpha$ is the fine structure constant, $\boldsymbol{\alpha}$ is the vector of Dirac matrices, $\omega$ stays for the incident and scattered photons energies, and $E_i$, $E_f$, and $E_a$ define the energies of the initial, final, and intermediate states, respectively. The quantum numbers $m_i$ and $m_f$ are the total angular momentum projections of the initial and final states, respectively. Summing over $a$ means that we take into account the contributions of all possible intermediate states instead of just the dominant one. One may notice that if the photon energy $\omega$ is close to $E_a - E_i$, there is a singularity in the first term in Equation (6). This singularity is exactly the case of the resonant intermediate state. In order to regularize the expression, we introduce the corresponding width $\Gamma_a$ in the denominator, i.e., we replace $E_a$ with $E_a - i\Gamma_a/2$ in the vicinity of the resonance [28,29]. This prescription corresponds to performing an infinite resummation of the radiative correction insertions for the resonant level [30,31].

Angular properties can be most conveniently described with the help of the multipole expansion of the transition operator [32]. Transition operators inside the matrix elements take the following form:

$$\boldsymbol{\alpha} \cdot \boldsymbol{\epsilon}_\lambda e^{i\mathbf{k} \cdot \mathbf{r}} = \sqrt{2\pi} \sum_{pJM} i^J \sqrt{[J]} (i\lambda)^p D^J_{M\lambda}(\varphi_{\mathbf{k}}, \vartheta_{\mathbf{k}}, 0) \boldsymbol{\alpha} \cdot \mathbf{a}^{(p)}_{JM}(\omega, \mathbf{r}), \tag{7}$$

where $\mathbf{a}^{(p)}_{JM}(\omega, \mathbf{r})$ denotes the electric ($p = 1$) and magnetic ($p = 0$) multipole components of the electromagnetic field, $J$ represents the photon angular momentum, and $M$ stands for its projection. The electric and magnetic multipole field components are defined as:

$$\mathbf{a}^{(0)}_{JM}(\omega, \mathbf{r}) = j_J(\omega r) \mathbf{Y}^M_{JJ}(\hat{\mathbf{r}}), \tag{8}$$

$$\mathbf{a}^{(1)}_{JM}(\omega, \mathbf{r}) = j_{J-1}(\omega r) \sqrt{\frac{J+1}{2J+1}} \mathbf{Y}^M_{JJ-1}(\hat{\mathbf{r}}) - j_{J+1}(\omega r) \sqrt{\frac{J}{2J+1}} \mathbf{Y}^M_{JJ+1}(\hat{\mathbf{r}}). \tag{9}$$

In the above equations, $\hat{\mathbf{r}}$ is the $\mathbf{r}$ unit vector, $j_J(\omega r)$ are spherical Bessel functions of order $J$, and $\mathbf{Y}^M_{JL}(\hat{\mathbf{r}})$ are vector spherical harmonics, defined by:

$$\mathbf{Y}^M_{JL}(\hat{\mathbf{r}}) = \sum_{m\mu} \langle Lm1\mu|JM\rangle Y_{Lm}(\hat{\mathbf{r}}) \boldsymbol{\chi}_\mu, \tag{10}$$

with Clebsch–Gordan coefficients $\langle Lm1\mu|JM\rangle$, ordinary spherical harmonics $Y_{Lm}(\hat{\mathbf{r}})$, and complex spherical unit vectors $\boldsymbol{\chi}_\mu$ expressed via Cartesian unit vectors as:

$$\boldsymbol{\chi}_\mu = \begin{cases} \dfrac{1}{\sqrt{2}}(\hat{\mathbf{x}} - i\hat{\mathbf{y}}) & \text{for } \mu = -1, \\ \hat{\mathbf{z}} & \text{for } \mu = 0, \\ -\dfrac{1}{\sqrt{2}}(\hat{\mathbf{x}} + i\hat{\mathbf{y}}) & \text{for } \mu = 1. \end{cases} \tag{11}$$

After the expansion (7) is applied, it is obvious that the angular distribution of the emitted radiation is fully defined by the Wigner rotation $D$-matrices $D^J_{M\lambda}(\varphi_{\mathbf{k}}, \vartheta_{\mathbf{k}}, 0)$. As we agreed to the $z$-axis being coincident with the propagation direction of the incident photon, all the angles included in the corresponding $D$-matrix are equal to zero and $D^J_{M\lambda_i}(0, 0, 0) = \delta_{M\lambda_i}$ [33]. Therefore, the angular dependence of the scattering amplitude (6) is defined by a single rotation matrix $D^J_{M\lambda_f}(\varphi, \vartheta, 0)$ with $\varphi$ and $\vartheta$ defined as shown in Figure 1. From now on, for brevity, we omit the arguments of the rotation matrix and always assume them to be $(\varphi, \vartheta, 0)$.

Now, after we write the transition operator in the multipole expansion (7), we can employ the angular momentum theory to deal with the matrix elements effectively. The sum over $a$ in Equation (6) implies not only summation over different states (with different principal quantum numbers $n$ and total angular momentum $j_n$), but also summation over all possible magnetic substates. These substates have different values of projection $m_n$ of the angular momentum $j_n$. A common way to simplify calculations with such a summation is to make use of the Wigner–Eckart theorem:

$$\langle n_2 j_2 m_2 | \boldsymbol{\alpha} \cdot \mathbf{a}^{(p)}_{JM} | n_1 j_1 m_1 \rangle = (-1)^{2J} \langle j_1 m_1 J M | j_2 m_2 \rangle \frac{\langle n_2 j_2 || \boldsymbol{\alpha} \cdot \mathbf{a}^{(p)}_J || n_1 j_1 \rangle}{\sqrt{2 j_2 + 1}} \,. \tag{12}$$

Applying the Wigner–Eckart theorem (12) together with the multipole expansion (7) to Equation (6), we come to the following form of the scattering amplitude:

$$
\begin{aligned}
\mathcal{M}(m_i, m_f, \lambda_i, \lambda_f) = {} & 2\pi \tfrac{\alpha}{\omega} \sum_{n j_n m_n} \sum_{\substack{J_1 J_2 \\ M_1 M_2}} \sum_{p_1 p_2} (-i)^{J_1 - J_2} (i\lambda_i)^{p_1} (-i\lambda_f)^{p_2} \sqrt{[J_1][J_2]} D^{J_2*}_{M_2 \lambda_f} \delta_{\lambda_i M_1} \\
& \times \left[ \frac{\langle j_n m_n J_2 M_2 | j_f m_f \rangle \langle j_i m_i J_1 M_1 | j_n m_n \rangle}{[j_n]^2} \frac{\langle f || \boldsymbol{\alpha} \cdot \mathbf{a}^{(p_2)}_{J_2} || n j_n \rangle \langle n j_n || \boldsymbol{\alpha} \cdot \mathbf{a}^{(p_1)}_{J_1} || i \rangle}{E_i + \omega - E_{n j_n}} \right. \\
& \left. + \frac{\langle j_n m_n J_1 M_1 | j_f m_f \rangle \langle j_i m_i J_2 M_2 | j_n m_n \rangle}{\sqrt{[j_f][j_i]}} \frac{\langle f || \boldsymbol{\alpha} \cdot \mathbf{a}^{(p_1)}_{J_1} || n j_n \rangle \langle n j_n || \boldsymbol{\alpha} \cdot \mathbf{a}^{(p_2)}_{J_2} || i \rangle}{E_f - \omega - E_{n j_n}} \right]
\end{aligned}
\tag{13}
$$

with $[J] = 2J + 1$. All the angle dependence is expressed now by the Wigner $D$-matrix, which depends on the indices $J_2, M_2, \lambda_f$. The function can be effectively dealt with using the angular momentum algebra apparatus.

### 2.3. Exact Calculations

In order to come now to the observables, we rewrite Equation (2) in terms of the matrix elements of the scattering operator in the following way:

$$\langle \lambda_f | \hat{\rho}_f | \lambda'_f \rangle = \frac{1}{\sqrt{2 j_i + 1}} \sum_{\lambda_i \lambda'_i} \sum_{m_i m_f} \mathcal{M}(m_i, m_f, \lambda_i, \lambda_f) \mathcal{M}^*(m_i, m_f, \lambda'_i, \lambda'_f) \langle \lambda_i | \hat{\rho}_i | \lambda'_i \rangle \,. \tag{14}$$

Here, we assume that the ion is initially unpolarized, and its polarization in the final state remains unobserved. The next part of the derivation is quite tedious, so we will only mention the operations that need to be performed. First, the product of two transition matrix amplitudes $\mathcal{M}\mathcal{M}^*$ in Equation (14) results in a product of two rotation matrices, as follows from Equation (13). This product can be written in terms of a double sum of a single rotation matrix multiplied by two Clebsch–Gordan coefficients. Next, we exploit formulas for the sums of products of Clebsch–Gordan coefficients (to sum over $m_i, m_f, M_1, M'_1$) and leave only the factors able to reveal some features of the angular distribution. Moreover, for the photon energies much smaller than the electron rest-mass energy, the second term in the square brackets of Equation (13) is much smaller than the first one. For that reason, we write down explicitly only those terms of Equation (14), which include only terms

originated from the first term. We band all the other terms together into a single symbolic notation and arrive at the following look of Equation (14):

$$
\begin{aligned}
\langle \lambda_f | \hat{\rho}_f | \lambda'_f \rangle \quad \sim \quad & \sum_{nj_n n'j'_n} \left[ \sum_{\lambda_i \lambda'_i} \sum_{KMN} \sum_{\substack{J_1 J_2 \\ J'_1 J'_2}} \sum_{\substack{p_1 p_2 \\ p'_1 p'_2}} \begin{Bmatrix} J_2 & j_f & j_n \\ j'_n & K & J'_2 \end{Bmatrix} \begin{Bmatrix} j_n & j_i & J_1 \\ J'_1 & K & j'_n \end{Bmatrix} (i\lambda_i)^{p_1} (-i\lambda_f)^{p_2} (-i\lambda'_i)^{p'_1} (i\lambda'_f)^{p'_2} \right. \\
& \times \quad (-1)^{j'_n - K} (-i)^{J_1 - J_2 - J'_1 + J'_2} \sqrt{[J_1][J_2][J'_1][J'_2]} D^K_{MN} \langle J_2 - \lambda_f J'_2 \lambda'_f | KN \rangle \langle J_1 \lambda_i J'_1 - \lambda'_i | K - M \rangle \\
& \times \quad \left. \frac{\langle f \| \boldsymbol{\alpha} \cdot \mathbf{a}^{(p_2)}_{J_2} \| nj_n \rangle \langle nj_n \| \boldsymbol{\alpha} \cdot \mathbf{a}^{(p_1)}_{J_1} \| i \rangle}{E_i + \omega - E_{nj_n}} \frac{\langle f \| \boldsymbol{\alpha} \cdot \mathbf{a}^{(p'_2)}_{J'_2} \| n'j'_n \rangle \langle n'j'_n \| \boldsymbol{\alpha} \cdot \mathbf{a}^{(p'_1)}_{J'_1} \| i \rangle}{E_f + \omega - E_{n'j'_n}} \right] \langle \lambda_i | \hat{\rho}_i | \lambda'_i \rangle \\
& + \quad O \left( \frac{E_i + \omega - E_{nj_n(n'j'_n)}}{E_f - \omega - E_{nj_n(n'j'_n)}} \right) .
\end{aligned}
$$
(15)

In the above equation, $K$ symbolizes angular momentum transferred to the system as a result of its interaction with two photons and indices $M, N$ effectively play the roles of its projections.

We conclude this section with a short remark concerning the numerics used in this work. As has been mentioned above, the calculation procedure implies a sum over the complete Dirac spectrum. In the numerical calculations, the entire spectrum is typically substituted by a finite set of discrete pseudostates (see, e.g., [34,35]). These pseudostates are then treated in the same manner as the bound states. Setting up the pseudostates requires a set of basis functions. One of the most popular choices of such a basis is a set of splines (piecewise polynomials), as they provide many advantages when it comes to numerical calculations [36]. Within the class of piecewise polynomials, B-splines (for details, see [34]) are especially well suited for atomic physics numerical tasks [35,37], and that is why they are used here as well. In the works [11,38], the B-splines basis set was also employed for the calculation of the Rayleigh scattering.

## 2.4. Relativistic Resonant Electric-Dipole Approximation

Equation (15) is not restricted to any particular kind of transition or certain states. However, when the photon energy is very close to the transition energy into a particular intermediate state (resonant state), the dominant contribution in the sum over $nj_n$ in Equation (15) is given by this state. Moreover, the restriction to the electric dipole transitions only allows us to obtain simple analytical expressions as for the angle-differential cross-section, as well as for the polarization. Under these assumptions, we arrive at the relativistic resonant electric-dipole approximation. To obtain this approximation from Equation (15), we restrict first the sum over $n, j_n, n', j'_n$ into the resonant state only. In the second place, we neglect the terms denoted by $O(...)$ and imply the electric-dipole approximation $J_1 = J_2 = J'_1 = J'_2 = 1$, as well as $p_1 = p_2 = p'_1 = p'_2 = 1$. With all these changes, we obtain:

$$
\begin{aligned}
\langle \lambda_f | \hat{\rho}_f | \lambda'_f \rangle^{\text{rel. resonant } E1} \quad \sim \quad & \sum_{\lambda_i \lambda'_i} \sum_{KMN} \begin{Bmatrix} 1 & j_f & j_n \\ j_n & K & 1 \end{Bmatrix} \begin{Bmatrix} j_n & j_i & 1 \\ 1 & K & j_n \end{Bmatrix} \lambda_i \lambda'_i \lambda_f \lambda'_f (-1)^{j_n - K} D^K_{MN} \\
& \times \quad \langle 1 - \lambda_f 1 \lambda'_f | KN \rangle \langle 1 \lambda_i 1 - \lambda'_i | K - M \rangle \frac{\left| \langle f \| \boldsymbol{\alpha} \cdot \mathbf{a}^{(1)}_1 \| nj_n \rangle \right|^4}{(E_f + \omega - E_{nj_n})^2 + \Gamma^2_{nj_n}/4} .
\end{aligned}
$$
(16)

For the calculations of the cross-section and degree of polarization of the scattered light, we have to calculate the trace or non-diagonal matrix elements of the resulting density matrix; see Equations (4) and (5). For this, we have to specify the total angular momentum of the intermediate (resonant) state $j_n$. Here, we consider $p_{1/2}$ and $p_{3/2}$ states. The first one, the $p_{1/2}$ state, results in the following angle-differential cross-section:

$$
\frac{d\sigma^{\text{rel}, p_{1/2}}}{d\Omega} (\text{circ. pol.}) \sim \frac{d\sigma^{\text{rel}, p_{1/2}}}{d\Omega} (\text{lin. pol.}) \sim 1
$$
(17)

and the linear polarization Stokes parameters:

$$P_1^{\mathrm{rel},p_{1/2}}(\mathrm{circ.\,pol.}) = P_1^{\mathrm{rel},p_{1/2}}(\mathrm{lin.\,pol.}) = 0\,, \qquad P_2^{\mathrm{rel},p_{1/2}}(\mathrm{circ.\,pol.}) = P_2^{\mathrm{rel},p_{1/2}}(\mathrm{lin.\,pol.}) = 0 \tag{18}$$

for circular (circ. pol.) and linear (lin. pol.) polarizations of the incident light. These results can be understood by the following argumentation. For the intermediate state with $j_n = 1/2$ 6j-symbols in Equation (16), ensure that $K = 0, 1$. Because of the symmetry of the Clebsch–Gordan coefficients, only even values of $K$ will survive, which leaves us with $K = 0$ and $D_{00}^0 = 1$. Since only the $D$-matrix defines the angular distribution, for this intermediate state, all scattering directions and linear polarization states are equally possible.

For the second one, the $p_{3/2}$ state, we start again with Equation (16) and see that $K = 2$ is now allowed. That means that a second-order Legendre polynomial is appearing in the angular distribution. Derivation results in the following angle-differential cross-section:

$$\frac{d\sigma^{\mathrm{rel},p_{3/2}}}{d\Omega}(\mathrm{circ.\,pol.}) \sim D_{00}^0 + \frac{1}{4}D_{00}^2 \sim 1 + \frac{3}{7}\cos^2\vartheta \tag{19}$$

and the linear polarization Stokes parameters:

$$P_1^{\mathrm{rel},p_{3/2}}(\mathrm{circ.\,pol.}) = -\frac{\sin^2\vartheta}{7/3 + \cos^2\vartheta}\,, \qquad P_2^{\mathrm{rel},p_{3/2}}(\mathrm{circ.\,pol.}) = 0 \tag{20}$$

for the case of circular polarization of the incident light.

As for the linear polarization case, we obtain:

$$\frac{d\sigma^{\mathrm{rel},p_{3/2}}}{d\Omega}(\mathrm{lin.\,pol.}) \sim 1 - \frac{3}{5}\cos^2\varphi\sin^2\vartheta \tag{21}$$

and:

$$P_1^{\mathrm{rel},p_{3/2}}(\mathrm{lin.\,pol.}) = -\frac{\sin^2\varphi - \cos^2\varphi\cos^2\vartheta}{5/3 - \cos^2\varphi\sin^2\vartheta}\,, \qquad P_2^{\mathrm{rel},p_{3/2}}(\mathrm{lin.\,pol.}) = \frac{2\sin\varphi\cos\varphi\cos\vartheta}{5/3 - \cos^2\varphi\sin^2\vartheta}\,. \tag{22}$$

*2.5. Nonrelativistic Resonant Electric Dipole Approximation*

It is well established by now that the nonrelativistic approximation can be employed to obtain rough estimations of the scattering processes. However, in many cases, it fails to describe the experimental results, even qualitatively [39–41]. Since the nonrelativistic resonant approximation has been used in the original Gamma Factory proposal [22] (see also [42]), here, we are going to work out the details of this approximation and compare it with previously outlined approaches.

In nonrelativistic treatment $2p_{1/2}$ and $2p_{3/2}$ states are considered as one degenerate $2p$ state. We can formally come to this type of consideration from Equation (16) if we characterize electron states with orbital quantum number $l$ instead of total angular momentum quantum number $j$. For the chosen states, that means the following substitutions: $j_{i,f} \to l_{i,f} = 0$ and $j_{n,n'} \to l_{n,n'} = 1$. After that, we deal with the cross-section and polarization in the same manner as before and obtain well-known expressions:

$$\frac{d\sigma^{\mathrm{nonrel}}}{d\Omega}(\mathrm{circ.\,pol.}) \sim 1 + \cos^2\vartheta\,, \tag{23}$$

$$P_1^{\mathrm{nonrel}}(\mathrm{circ.\,pol.}) = -\frac{\sin^2\vartheta}{1 + \cos^2\vartheta}\,, \qquad P_2^{\mathrm{nonrel}}(\mathrm{circ.\,pol.}) = 0\,, \tag{24}$$

$$\frac{d\sigma^{\mathrm{nonrel}}}{d\Omega}(\mathrm{lin.\,pol.}) \sim \sin^2\varphi + \cos^2\varphi\cos^2\vartheta\,, \tag{25}$$

$$P_1^{\text{nonrel}}(\text{lin. pol.}) = -\frac{\sin^2\varphi - \cos^2\varphi\cos^2\vartheta}{\sin^2\varphi + \cos^2\varphi\cos^2\vartheta}, \qquad P_2^{\text{nonrel}}(\text{lin. pol.}) = \frac{2\sin\varphi\cos\varphi\cos\vartheta}{\sin^2\varphi + \cos^2\varphi\cos^2\vartheta}. \qquad (26)$$

These expressions can be also found in [6,43,44].

## 3. Results and Discussion

### 3.1. Ion Rest Frame

We perform numerical calculations for both initial circular and initial linear polarization. Moreover, to investigate the limits of the resonant approximation, we present the results for two different photon energies: at and off the resonance. In the "resonance" scenario, we set the photon energy to match the corresponding transition energy. In the "off-resonance" case, the photon energy is chosen to be $30\Gamma_{2p_{1/2}}$ or $30\Gamma_{2p_{3/2}}$ above the corresponding resonances $2p_{1/2}$ or $2p_{3/2}$, respectively. For the case of the H-like lead ion, these widths take the following values $\Gamma_{2p_{1/2}} = 19.4$ eV and $\Gamma_{2p_{3/2}} = 16.9$ eV. The resonance and off-resonance scenarios lead to essentially different values of the cross-section. Since we are mostly interested in the form of the angular distribution, we normalize the cross-section results for all scenarios and compare the line shapes. Here, we note that the shapes obtained via the nonrelativistic and relativistic resonant electric-dipole approximations are independent of the photon energy and are given by Equations (23)–(26) and (17)–(22), respectively.

Figure 2 shows the cross-section and polarization angular distributions in the case when the incident light is circularly polarized. For the $2p_{1/2}$ intermediate resonance (left column), we see that for the "resonance" photon energy, the relativistic resonant electric-dipole approximation (black dashed line) almost coincides with the exact calculations (solid blue line). The results differ for less than 0.3% of the mean value, and this difference is not visible in the figure. We can also see that both cross-section and polarization distributions are flat, as expected form Equations (17) and (18). Since the nondipole photon-ion interaction has no effect on the $|2p_{1/2}\rangle \rightarrow |1s_{1/2}\rangle$ transition, the $2p_{1/2}$ intermediate state is a perfect candidate for the demonstration of an effect beyond the resonant approximation. One can see that in the "off-resonance" case, the angle-differential cross-section deviates from the flat line by around 20%. At the same time, the degree of linear polarization reaches 0.25 for $\vartheta = 90°$. The middle column in Figure 2 shows the results for the $2p_{3/2}$ resonant state. First of all, we notice that the shape of the cross-section distribution obtained by the exact calculations differs significantly from the relativistic resonant electric-dipole approximation, Equations (19) and (20). Comparing the cross-section distributions in the "resonance" and "off-resonance" cases, we see that even that far from the resonance (30 line widths), the effect of nondipole contributions is more pronounced. On the other hand, the polarization seems to be more sensitive to the contributions from nonresonant states. When we consider $2p$ as the intermediate state, we imply nonrelativistic approximation. In this approximation, the line shapes of both cross-section and polarization distributions do not change with changing the photon energy and agree with Equations (23) and (24).

In Figure 3, we present the results for the linearly polarized incident light. When we switch from circular to linear polarization, the system loses circular symmetry in the $xy$-plane. That means that the results are different for different values of azimuthal observation angle $\varphi$. Here, we consider only the case of $\varphi = 0°$. The discussion of the results would be quite similar to the previous figure, so we only make several observations: (i) for the $2p_{1/2}$ resonant state, we see a similar picture as in Figure 2, except that the effect of the nonresonant states is even more pronounced; (ii) for the $2p_{3/2}$ resonant state, we note that the results in the relativistic resonant electric-dipole approximation follow Equation (21) and Equation (22); one can also see that both distributions, in this case, vary significantly more strongly than for the circular polarization; (iii) for the nonrelativistic case, the shapes of both lines are essentially different as compared to the $2p_{1/2}$ and $2p_{3/2}$ cases. In accordance with Equations (25) and (26), they are also different from the case of circular initial polarization.

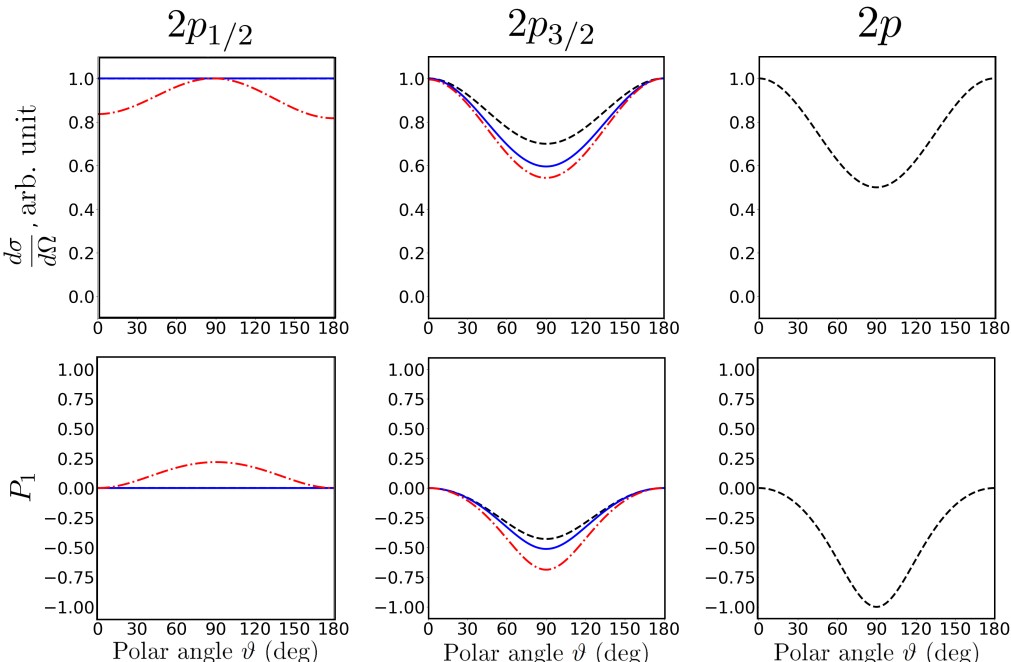

**Figure 2.** Cross-section and polarization angular distribution for different resonant states: $2p_{1/2}$—left column, $2p_{3/2}$—middle column, and nonrelativistic $2p$ state—right column. Different approaches are compared: dashed black line—relativistic resonant electric dipole approximation, dashed-dotted red line—exact calculations for the "off-resonance" scenario (see the text), solid blue line—exact calculations at the corresponding resonance. Incident light is assumed to be circularly polarized.

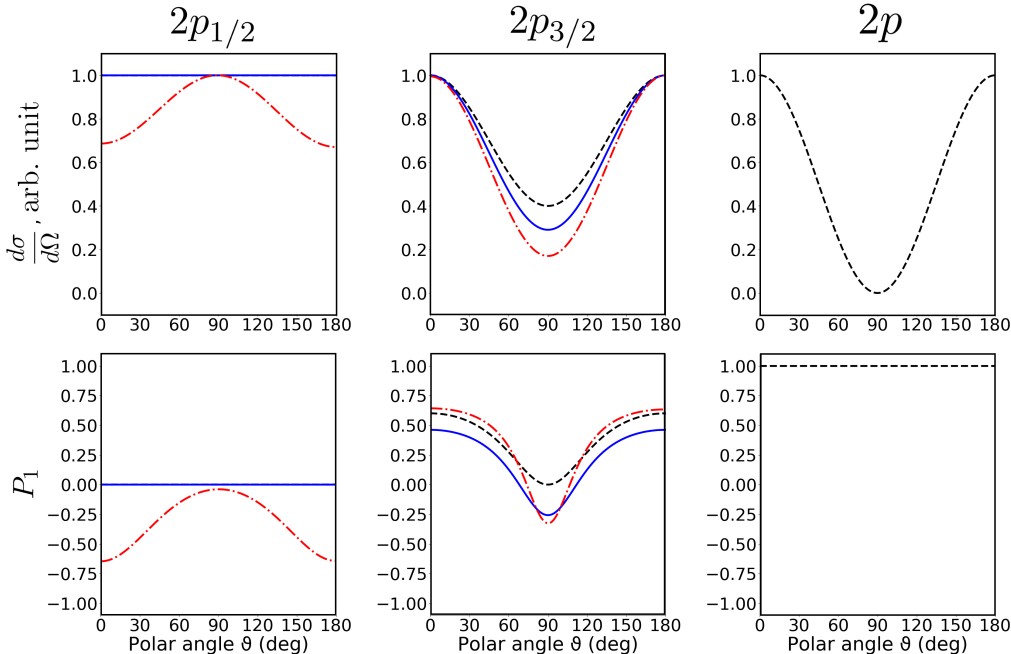

**Figure 3.** the same as Figure 2, but for the linearly polarized incident light and for the azimuthal observation angle $\varphi = 0°$.

The large detuning of the photon energy in the "off-resonance" scenario is a limiting case and considered here for studies of the importance of the nonresonant state's contribution. For the

smaller values of the detuning, the nonresonant state's contribution is lower. Specifically, in all the cases considered above for photon energies in the range of $5\Gamma_{2p_{1/2}}$ or $5\Gamma_{2p_{3/2}}$ from the resonance, the results of the relativistic resonant electric-dipole approximation differ from exact calculations for less than 12%. Therefore, we expect that this relativistic approximation performs well enough for the design of experiments [22], in contrast to the presently employed nonrelativistic one.

### 3.2. Laboratory Reference Frame

Since the primary application of the results obtained in this work is expected to be the Gamma Factory initiative, we make two additional points. The first one is that the results presented above can be transformed into the laboratory reference frame for any given value of Lorentz factor $\gamma$. Assuming that the incident photon and the ion move towards each other (so, the ion also moves along the $z$-axis), this can be done by combining any angular distribution with the well-known relativistic aberration formula:

$$\cos \vartheta' = \frac{\cos \vartheta - \beta}{1 - \beta \cos \vartheta},$$

$$(27)$$

where $\vartheta'$ is the scattering angle in the laboratory reference frame counted from the $z$-axis and $\beta$ is the ion velocity relative to the speed of light. The latter can be expressed via $\gamma$ from:

$$\gamma = \frac{1}{\sqrt{1 - \beta^2}}.$$

$$(28)$$

In this head-on collision scenario, the azimuthal angle $\varphi$ transforms trivially to $\varphi'$, i.e., $\varphi' = \varphi$. Corresponding calculations are presented in Figure 4 for the case of circular initial polarization and $2p_{3/2}$ taken as the intermediate state. One can see from the figure that with growing $\gamma$, the distribution is concentrated in the vicinity of $\vartheta' = \pi$. For a fixed high value of $\gamma$, one obtains a well-known narrow cone-like radiation distribution, which is characteristic for relativistically moving particles. Moreover, as we discussed in the previous subsection, the relativistic resonant electric-dipole approximation provides a reasonable estimation of the angle distributions. It means that for the design of experiments, one can transform to the laboratory frame the distributions given by Equations (17)–(22).

The second point is that, although we so far considered a single process (meaning that we have just one scattered photon), in the experimental case, one has a full beam of photons. Then, the value of the differential cross-section in a certain direction describes the number of photons emitted in that direction. Similarly, the value of $P_l(\vartheta')$ describes the polarization state of the photon beam in the direction of $\vartheta'$. Because of the concentration of the radiation in the direction of $\vartheta' = \pi$, all the photons at different polarization states form a single beam. The polarization properties of this beam are still to be defined. The recipe for how to define them will be discussed elsewhere, but we point out the obvious importance of the consideration of the polarization distribution for studying this issue.

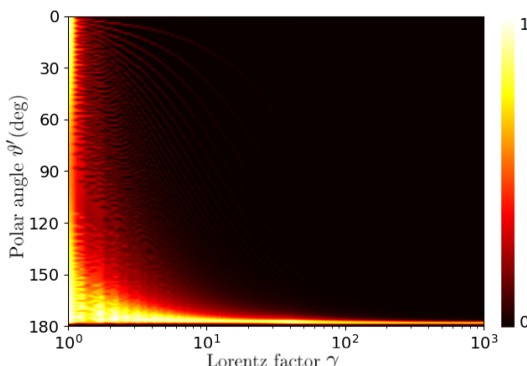

**Figure 4.** Angular distribution of the scattered light in the laboratory reference frame at different values of the Lorentz factor. The color illustrates the values of the cross-section, normalized to one for a fixed value of $\gamma$.

## 4. Conclusions

In this work, we investigated the excitation and subsequent decay of the $2p$ electron states of the H-like lead ion using the well-developed theoretical apparatus of scattering theory. The investigation included the angular distribution of the cross-section and the degree of linear polarization of the scattered light. We derived the exact expression for density matrix elements, which were necessary for the calculation of these two observables. The expressions were explicitly tailored for two resonant electric-dipole approximations: nonrelativistic and relativistic. We performed numerical calculations with the exact formula, as well as with both approximations. The results were compared for different intermediate (resonant) states $2p_{1/2}$ and $2p_{3/2}$, as well as for different photon energies with respect to the resonance. The findings showed two main points. First, in most cases, the results of the nonrelativistic approach were substantially different from the exact calculations. Second, for moderate detuning of the photon energy from the resonance transition energy, the relativistic resonant electric-dipole approximation provided accuracy around 10% for all considered scenarios. The analytical expressions derived in this approximation were quite general and could be used for other types of H-like ions, as well. Therefore, Equations (17), (19), and (21) derived for the cross-section and Equations (18), (20), and (22) for polarization could be used in the design of near-resonance Rayleigh scattering experiments and the interpretation of the results. In particular, we expect these formulas to be favorable to theoretical investigations at Gamma Factory. Additionally, it was demonstrated that the derived analytical expressions in the relativistic resonant electric-dipole approximation could be easily transformed into the laboratory reference frame.

**Author Contributions:** Investigation, D.S.; methodology, A.V.V.; supervision, S.F.; writing, original draft, D.S.; writing, review and editing, A.V.V. and S.F. All authors have read and agreed to the published version of the manuscript.

**Funding:** This research received no external funding.

**Conflicts of Interest:** The authors declare no conflict of interest.

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
