# Peer review of "Elastic Photon Scattering on Hydrogenic Atoms near Resonances"

_atoms, doi:10.3390/atoms8020012_

Round 1

Reviewer 1 Report

Report on the Ms ID Atoms-747171

The manuscript is dealing with the elastic photon scattering on hydrogenic atoms near resonances, as stated in the title. The formalism is presented in details and the relatively new aspect of it is the discussion of the effects due to near-resonances channels.
The findings are expected (relativistic descriptions are given correct
results). I think the manuscript could be published as it is perhaps with the addition in the abstract that the article is devoting a sizeable part of it to a sort of pedagogical review of the formalism.

Reviewer 2 Report

The manuscript under review is devoted to the theoretical investigation of the interaction of ion beams with light, an interesting topic that resides within the scope of Atoms.

The work presents an increased understanding of the elastic light scattering on the heavy highly ionized hydrogen-like atoms, being focused on the particular case of Pb81+ ion and photon energies in the vicinity of 1s-2p1/2 and 1s-2p3/2 resonances.

In my opinion, the conducted research constitutes a significant contribution to the field of theoretical atomic physics and spectroscopy.

The title clearly identifies the subject matter of the article.

Most of the abstract is succinct and comprehensible to a non-specialist. However, the Authors are recommended to refer to the nature of the "near-resonance regime" — i.e., to relate it to the (specific) photon energy - electron transition resonance at first mentioning in line 3. As well, it would be better to replace QED by quantum electrodynamics, bearing in mind the exposure of the Abstract to a very wide audience and the most widespread meaning of QED (Quod Erat Demonstrandum).

The keywords selected by the Authors are appropriate for indexing the article.

The content presented in the manuscript appears to be sound. However, the Authors must add further discussion on its significance. The ion Pb81+ — a lead atom that has been stripped away all its electrons except one — is rather an exotic object and its bringing into the scene immediately raises questions like How frequently it appears in the experiments? How easy the corresponding ion beam can be generated, presently or in the future, if it possesses some useful properties? In a more general perspective: Why the preference is given to heavy H-like ions? Why the lead is chosen among possible case studies? In the article, the Authors refer the reader to the Gamma Factory initiative, but I do believe that it is very important to provide at least brief clarification directly in the text.

The quality of English language usage and grammar is, in general, appropriate for Atoms. Moderate English changes are advisable, in particular, with respect to the comments related to the presentation quality listed below:

Lines 55-58

  • The project, which is perusing building at CERN (European Organization for Nuclear Research) a γ-ray source with photon energies several orders of magnitude higher than available nowadays anywhere in the world with beam intensities comparable with those produced by free-electron lasers.

Please rephrase this compound subordinate sentence, which looks fragmented as the main clause is reduced to "The project".

Line 61

  • here, the incident photon...

The more common notion implies either starting a new sentence (Here, the incident photon...) or using where without separating comma (where the incident photon...).

Lines 67-68

  • The scattered photon propagates in the direction of kf along the z'-axis.

Why? Did the Authors mean that the direction of the scattered photon propagation is denoted as the z'-axis?

Unnumbered line above Eq. (4)

  • Keeping this in mind the angle-differential cross section is given by the trace of the density matrix of the scattered photon:

Please provide a reference supporting this statement: the wide Atoms readership may not be very familiar with "nuts and bolts" of the density matrix formalism.

Eq. (6)

Is the use of the curly brackets {·} meaningful? If not, why don't the Authors use (·)?

Line 108

  • In the above equation, J symbolize angular momentum transferred to the system...

Earlier the same variable J was introduced as "the photon angular momentum", see the second line below Eq. (7). Can you avoid confusion using a different notion or style for one of the occurrences?

General remarks about the Conclusion section

  • It is not completely clear how the Authors are going to apply the obtained results within the scope of the Gamma Factory initiative or in a wider context.
  • Do all the conclusions made can be applied to any H-like ion or there are some results specific to the lead ion?
  • Figure 4 contains a color-coded plot. The Authors state that "the color bar illustrates the values of cross section", but no color bar with numerical reference is given. In the plot, there are flame-like horizontal strips and vertical wave-like disturbances. Is it a numerical effect or there is some physics behind?
